# Combining Azimuthal and Polar Angle Resolved Shadow Mask Deposition and Nanosphere Lithography to Uncover Unique Nano-Crystals

**DOI:** 10.3390/nano12193464

**Published:** 2022-10-04

**Authors:** Arnab Ganguly, Gobind Das

**Affiliations:** Department of Physics, Khalifa University, Abu Dhabi 12788, United Arab Emirates

**Keywords:** nanosphere lithography, angled shadow mask deposition, symmetry breaking, angular acceleration, nanopyramid

## Abstract

In this article, we present a systematic investigation on a multistep nanosphere lithography technique to uncover its potential in fabricating a wide range of two- and three-dimensional nanostructures. A tilted (polar angle) electron beam shower on a nanosphere mask results in an angled shadow mask deposition. The shape of the shadow also depends on the azimuthal angle of the mask sitting on top of the substrate. We performed angled shadow mask depositions with systematic variation of these two angular parameters, giving rise to complex nanostructures (down to 50 nm), repeated over a large area without defect. In this article, nanosphere lithography with two- and four-fold azimuthal symmetry was studied at constant tilt angles followed by variations in tilt without azimuthal rotation of the substrate. Finally, both angular parameters were simultaneously varied. The structure of shadow crystals was explained using Matlab simulation. This work stretches the horizons of nanosphere lithography, opening up new scopes in plasmonic and magnonic research.

## 1. Introduction

In nanotechnology, conventional lithography (CL) is a popular tool used for fabricating various nanostructures. However, in the case of photolithography, it is tricky to obtain well-resolved structures in the sub-micron range. E-beam lithography has higher resolution, but large-area patterning is not feasible with this technique. Moreover, CL is not suitable for patterning 3D nanostructures (with asymmetry along thickness). A 3D structure can be obtained using a focused ion beam, but it is also limited to large-area patterning. To cap it all, these techniques are expensive and sensitive to cleanroom conditioning; hence, they are not suitable for large-scale industrial applications. On the other hand, nanosphere lithography [1,2,3,4,5] (NSL) is a technique which has all the qualities required once it is combined with angled shadow mask deposition. It offers an easy alternative to realize large-area patterning of 2D and 3D nanocrystals in a cost-effective manner and does not necessarily require special cleanroom conditioning. The only bottleneck with NSL is that it has limited control on structures, which means the technique cannot create a wide range of nanostructures. It is conventionally used to create triangular crystals with a hexagonal close-packed (HCP) order [3,6,7]. There have been few efforts in the past to expand the scope of nanostructure fabrication using this technique [8,9]. For this reason, NSL does not have versatile applicability, despite its immense advantages over other techniques. To overcome this challenge, a lot of research efforts have been made for nanocrystal structures breaking HCP symmetry. A symmetrical breaking along the thickness by non-uniform etching was observed by Darvil et al. [8]. Myint et al. [9] observed crystal structures from a three-fold azimuthal symmetry deposition. In our work, we uncovered the crystal nanostructures obtained from two- and four-fold azimuthal rotations of the substrate at constant tilt angles, variable tilt angle without substrate rotation, and a simultaneous variation of the two. We observed large-area repetition of unique and complex unit cells different from the conventional HCP-like structures observed previously using a similar technique. In our study, nanostructures were obtained both in two and three dimensions. The formation of all complex nanostructures presented in this paper were interpreted and correlated by Matlab simulations. Additionally, unique 3D structures through angular integration were discussed in relevant contexts. Chiral spectroscopy and surface-enhanced Raman spectroscopy (i.e., plasmonic methods) are being used intensively for chemical, biological, and environmental monitoring applications. Studies have focused on unique metallic nanostructures with different symmetry, inter-structural gaps, and chirality for controlling electromagnetic hotspots and varying optical properties based on the orientation of structural elements [10,11,12,13,14,15]. Moreover, magnetic memory applications, such as data storage, data transfer, and logical devices rely on the controlled propagation of spin waves through magnonic crystals via exchange interactions. Over the last decade, research efforts have been invested in controlling properties of spin waves through variation in thickness and shape anisotropy of nanostructures [16,17,18,19,20,21]. Hence, periodic nanostructures, rich in various spin-wave modes, are in high demand. In our study, we fabricate nanocrystals of different shapes and sizes, with different inter-structural gaps and symmetry, along with special (continuous/step-like) variations in thickness, which are resourceful candidates for magnonics/plasmonics research. Our study brings the versatility of the NSL technique, in combination with angled shadow mask deposition, to the next level, opening up the scope for plasmonics [1,22,23,24,25,26] and magnetics [7,27,28,29,30].

## 2. Methods

### 2.1. Experimental

The study focuses on a scheme to fabricate a library of nanostructured arrays with varying shapes, sizes, symmetries, orientations, and the separation of the elements along with asymmetry along the thickness. To develop such structures, our experiment employs the NSL technique combined with a systematic variation in angular parameters during the shadow mask deposition of metal. The first step for NSL patterning is the deposition of the nanosphere mask on substrate. The process involves several optimizations of parameters through trial and error to achieve uniformity and lattice periodicity in structural patterning. With the process being stochastic, making a defect-free plasmonic lattice over a large surface area [4,31] is one of the biggest challenges that we have achieved using the recipe described in this work. The Si wafer is treated with Ar/O_2_ plasma etching to remove organic impurities on the substrate and make the surface hydrophilic [32,33]. We found that this treatment had a significant impact on the mobility of nanoparticle mask units, helping them spread over the surface of Si in a HCP lattice structure. Without plasma cleaning, the nanosphere masking process faces several difficulties, such as random agglomeration, multilayer formation, island formation, point defect (i.e., missing particle) dislocations, and uneven gaps between particles. In the second step, a solution of micro/nanobeads and ethanol is drop-casted on the substrate and allowed to be self-assembled in the HCP structure. The molar ratio of the solvent and the surface area of the substrate play a crucial role in the self-assembly process. The solvent provides mobility to the micro/nanobeads until it is evaporated. Once the solution is dried, the particles cease to move. The volume of the drop-casted solution and the surface area of the surface is carefully chosen so that particle density per unit square area is just sufficient to cover the entire surface with a monolayer particle mask. A higher concentration of micro/nanobeads would cause the formation of a multilayer mask and a smaller density would results in the formation of islands on the uncovered surface. External heating accelerates the evaporation process, allowing a smaller time window for particle dynamics to occur. This is specifically helpful when dislocation of particles from a previously arranged lattice is observed, due to long-time exposure for particle dynamics. Figure 1a shows a scanning electron microscope (SEM) image of a nanosphere mask arranged in HCP order. The inset shows the image at a higher magnification. Figure 1b shows a SEM image of the patterns of nanostructures through the normal deposition of metal after removal of the nanosphere beads. In our study, we used polyethyleneimine nanosphere beads of 2 µm, 500 nm, and 200 nm diameter from Sigma-Aldrich (St. Louis, MO, USA). However, in this paper, we present all experimental observations for only 2 µm to demonstrate clear and distinct nanostructures. In the next step, the Si substrate is attached to a slope in order to receive a tilted (polar angle *φ*) shower of metal with respect to the substrate during deposition, as shown in Figure 1c. A 25 nm thick Ti (from Kurt J. Lesker Company Ltd., East Sussex, UK) is deposited using an e-beam evaporator at a chamber pressure of 10^−3^ Pa. The deposition is repeated multiple times with a manually adjusted azimuthal angle (*θ*) and polar angle (*φ*), as defined in the schematic diagram shown in Figure 1c. After the deposition, the sample is ultrasonicated in isopropyl alcohol to remove the nanosphere masks along with the material deposited on them. The material grains that find the gaps of the nanosphere assembly in every step of shadow deposition remains a lithographed structure on the substrate. In our study, we demonstrate the variation of *φ* at a constant *θ*, creating step-like or continuous 3D structures, variation of *θ* at constant *φ* at different symmetries, and finally, chiral structures from simultaneous variation of *θ* and *φ*.

### 2.2. Simulation

A Matlab simulation was used to describe the structure formed through the combination of NSL and angled shadow mask deposition. A code was written to solve equations of straight lines and a set of spheres on a plane surface. The straight lines correspond with the showering of metal atoms in the e-beam evaporation chamber, the spheres correspond with the micro/nanobeads mask unit, and the plane corresponds with the substrate. The existence of a real solution between a sphere and straight line implies that the evaporated beam is hitting the nanosphere, whereas an imaginary solution indicates no contact between them. The first case corresponds to shadow, whereas the second case corresponds to deposition on the substrate. In the second case, the straight line and substrate plane is further solved for the coordinates of the deposition. Figure 1d shows the scheme of the simulation. The green dashed line corresponds with the beam going to deposit, whereas the green dots correspond with the location of deposition on the substrate. The red dashed line corresponds with the deposition on the mask responsible for the shadowing effect.

## 3. Results

### 3.1. Variation of Tilt without Azimuthal Rotation

In the most common e-beam evaporation configuration, the substrate is normal in the directional vector of the evaporation, i.e., the tilt angle *φ* becomes zero. In that case, if the substrate is azimuthally rotated to any arbitrary angle *θ*, the shadow will be exactly the same, which is nothing but the conventional HCP pattern of triangles. However, when the value of *φ* is something other than zero, the shadow will be varying for different *θ*. In other words, for *φ* = 0, the substrate is azimuthally symmetric, whereas it requires a nonzero value of *φ* (i.e., *φ* ≠ 0) for the symmetry breaking of *θ*. Hence, we can say that *φ* is the most important configurational parameter that needs to be investigated in order to obtain a wide range of nanostructures using the NSL technique. At a higher value of *φ,* the deposition vector is more oblique with respect to the substrate. Hence, a larger area of the substrate comes under the micro/nanobead shadow than the exposure, resulting in a pattern of smaller size. Thus, there will be no pattern after a certain *φ*, as the whole surface comes under the shadow of the nanosphere mask. We have found the critical angle, using Matlab, to be *φ* = 45°, above which there will be no pattern on the substrate (see Appendix A). Considering the practical constraints and resolution of the evaporation system, a window of confidence for *φ* was chosen to be from 10 to 35° for this study.

In this section, we discuss 3D nanostructures obtained from the variation of *φ* at constant *θ*, as shown in Figure 2. The inset in each figure is a magnified image of the unit cell of the lattice. Panel 1 of Figure 2a shows a SEM image of the four-step deposition at *θ* = 20° and *φ* = 10, 15, 20, and 25°. Four curved arm triangles of different sizes and aspect ratios were overlapped with one another in a gradual relative shift. Despite the same thickness at every step of the deposition, the cell structure is non-uniform along its thickness, due to the overlapping region. Thus, each deposition step acts as a building block to form vertically asymmetric 3D nanostructures. Panel 2 of Figure 2a include the simulated structures, which were consistent with experimental observations. Four depositions (D1 to D4) are indicated by the different colors mentioned in this figure. The same color convention is used for step deposition throughout the rest of the paper. The other colors observed in this figure are the linear sum of two or more colors in the color scale, occurring in the superposed region. This explains our experimentally observed step-like 3D structured arrays. In the case of a deposition being integrated over a continuously varying *φ* instead of only four depositions, we would obtain exciting uniform 3D unit-cells, rather than steps, as shown in Panel 3 of Figure 2a. The structure was simulated in Matlab by solving equations for *θ* = 20° and *φ* varying from 0 to 30° in steps of 1°. The color corresponds to the normalized height, as shown in the color bar. The same color bar is used for continuous deposition discussed throughout the rest of the paper. The lattice looks like a comet, which is actually a combination of a pyramid and a pentagonal-pyramid with asymmetric arms. Similarly, Figure 2b,c shows the results for *θ*
*=* 0° and *θ* = −45°. Panel 1 and Panel 2 (SEM and simulation images) of Figure 2b,c shows the triangles are shifting in different angles compared with Figure 2a, resulting in different 3D structures. The simulated structures were consistent with experimental findings. The *φ* integrated structure in Panel 3 of Figure 2b is a pair of nanopyramids each having a vertical plane, whereas Panel 3 of Figure 2c is a combination of Figure 2a,b Panel 3.

So far, we have discussed integration of *φ* corresponding with a continuously changing tilt of substrate at a constant angular velocity. Herein, we introduce a new control parameter called angular acceleration to modulate shapes of 3D lattice structures using NSL. In case of accelerated substrate rotation, the deposition time corresponding with each step of *φ* are unequal. At lower *φ*, the deposition time is longer, whereas for higher *φ* values, it is shorter, resulting in a thicker deposition at lower *φ* compared with higher *φ*. The shape also depends on the value of acceleration. Thus, the structures can be engineered by changing the equation of motion of the substrate. The effect of accelerated *φ* integration is studied on the nanocomet structure of Figure 2a. Figure 3a–c shows structures from positive, zero, and negative acceleration, which are significantly different from each other (Figure 3b is the same as Figure 2a Panel 3).

### 3.2. Variation of Azimuthal Angle at a Constant Tilt

#### 3.2.1. Two-Fold Symmetry

Figure 4 shows lattice structures obtained from two-fold azimuthal symmetry (*φ* is constant, *θ* = *θ_i_*, and *θ_i_* + 180°). Herein, Panel 1 presents the SEM images of experimentally obtained nanostructures and Panel 2 shows the simulated results. The results show the unique periodic patterns for different *θ*, keeping *φ* constant. Breaking the conventional hexagonal symmetry, the lattice can rather be explained by a line symmetry. For Figure 4a–f, the values of *φ* are constant at 15 and 25°, respectively. The two step deposition in each figure correspond to *θ* = *θ_i_* and *θ_i_* + 180°, with *θ_i_* being the *θ* for the first deposition. In Figure 4a–c as well as in Figure 4d–f, *θ_i_* varies as 0, 15, and 30°. In Figure 4a, pairs of triangles are oriented in a zigzag fashion around periodically occurring line gaps, whereas in Figure 4b,c, a combination of four triangles form the unit cell structures. The structures for *φ* = 25° are significantly different from those obtained from *φ* = 15°. In Figure 4d, we obtain square structures sliced by line gaps. In Figure 4e, a combination of squares and triangles are found, whereas Figure 4f is a combined array of stars and triangles. The consistency between experimental and simulated results explains the formation of the structures through the angled shadow mask deposition technique.

#### 3.2.2. Four-Fold Symmetry

In Figure 5, the nanostructures were produced using four-fold symmetry. It is a quantitative addition of two times the two-fold symmetry perpendicular to each other, resulting in *θ* = *θ_i_* and *θ_i_* + (90, 180, and 270°). Figure 5a,e depicts the SEM images for two different *θ_i_* (15 and 355°, respectively) at a constant *φ* of 25°. A bunch of eight triangles forming a unit cell makes the crystal structure more interesting. The orientation of triangles in Figure 5a,e are explained by the simulation results in Figure 5b–d and Figure 5f–h, respectively. Figure 5b,c and Figure 5f,g are the two two-fold symmetry results that have a relative phase δ*θ_i_* = 90°. The lattice structure of Figure 5b,c resembles Figure 4e, whereas Figure 5f,g resembles Figure 4d,f. Figure 5d,h shows the superposed image of Figure 5b,c,f,g, respectively, which are in a good agreement with the experimentally observed structures in Figure 5a,e. Figure 5i,j shows the experimental and corresponding simulation images from the two-step deposition (*φ* = 25°, *θ* = 0 and 90°). Unlike the symmetry structures discussed earlier, the structure is lacking symmetry in *θ*. However, it is the half of a four-fold symmetry or semi-four-fold symmetric which makes the crystal unique in this class. The angular parameters of Figure 5 are summarized in the following Table 1.

### 3.3. Simultaneous Variation of Azimuthal Angle Tilt

In Figure 6, four-fold deposition symmetries are obtained from a complex gyration of the sample, having simultaneously varying *θ* and *φ*. The value of the angular parameters for the four depositions are *θ* = *θ_i_*, and *θ_i_* + (90, 180, and 270°) and *φ* = 15, 18, 22, and 29°. The left panel of Figure 6a,b are the SEM images for *θ_i_* = 113 and 130°, respectively. The right panels are the corresponding simulation results. A set of eight triangles of different sizes form unit cell structures consistently repeated over the substrate. The unit cells from both the experiment and simulation are shown in the inset, which are consistent with one another. With every change in *θ_i_*, the triangles reshuffle their positions with respect to one another, due to the shifted shadow position, giving rise to new complex periodic nanostructures, as shown in the examples of Figure 6a,b.

In Figure 7, we discuss the simulated structure from integration over a full rotation of *θ*. The result obtained from a continuously varying *φ* is compared with a constant *φ* upon rotating *θ*. In the simulation, *θ* is varied from 0 to 359° in steps of 1°. For Figure 7a–c, the values of *φ* are kept constant at 15, 25, and 30°, whereas in Figure 7d, *φ* is continuously increasing from 15 to 30° at a constant angular velocity over the span of the full rotation of *θ*. The 3D structures in Figure 7a–c are distinct. The non-standard 3D structure in Figure 7a looks like a hollow nanovolcano with triangular opening with peaks at each corner. Figure 7b looks like rings interconnected with nanobridges, whereas Figure 7c can be best described by nanowalls on the alternate arms of a hexagonal gallery. From this result, it is clear that the shape of the 3D structure dramatically changes for different values of *φ*. The structure from continuously varying *θ* and *φ* is interesting, as shown in Figure 7d. It looks like connected nanospiral formations connected to each other.

## 4. Conclusions

In this article, the potential of NSL in combination with an angled shadow mask deposition is explored. A 3D rotation of substrate allows shadows of various shapes, inter-structural gaps, and symmetries, resulting in exciting crystal structures in 2D and 3D. A two- and four-fold symmetry rotation of the azimuthal angle (*θ*) with constant or variable polar angles (*φ*) were systematically studied. The study included development of nanostructures after e-beam metal evaporation, as well as a simulation experiment performed in Matlab. The two- and four-step deposition discussed in this paper were analytically interpolated to a larger number of steps to mimic a continuous rotation of substrate in the spherical coordinate system. The results obtained were periodic arrays of exciting 2D and 3D unconventional lattices brought into light for the first time in the scheme of nanosphere lithography. The formation of the nanostructures was well-interpreted by simulation results. We found that angular acceleration of the rotating substrate could also be used as a control parameter for modulating the structure of 3D structures. These structures could be of great importance in the field of plasmonics and magnonics. This study plays a prominent role in expanding the scope of the NSL fabrication technique, both in research and in industry.

## Figures and Tables

**Figure 1 nanomaterials-12-03464-f001:**
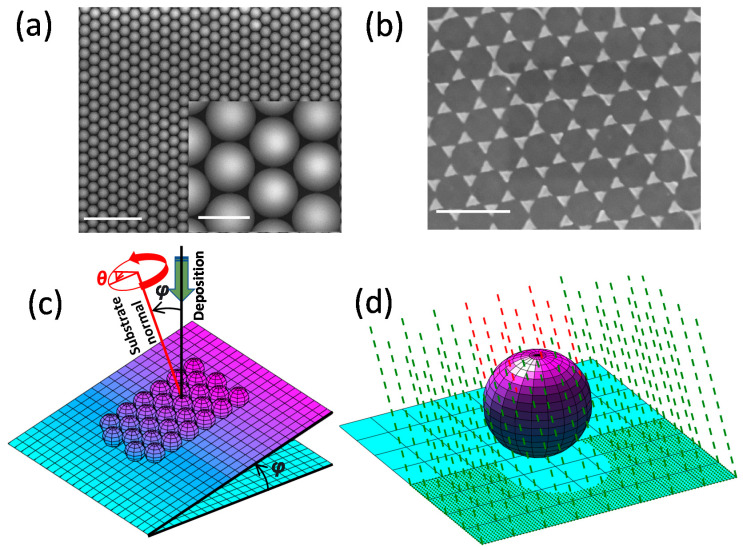
(**a**) Nanosphere mask on Si substrate. Scale bars of the image and the inset correspond to 10 µm and 2 µm, respectively. (**b**) NSL pattern for vertical deposition. Scalebar correspond to 4 µm. (**c**) Schematic diagram of the singular positioning of the sample for the e-beam deposition to attain the unique nanostructure, and (**d**) the simulation scheme for shadow deposition.

**Figure 2 nanomaterials-12-03464-f002:**
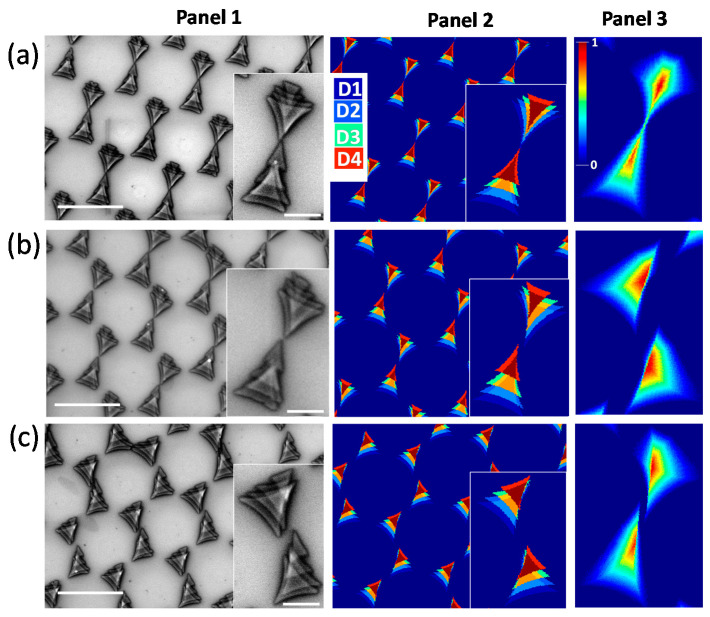
Four-step angled shadow mask deposition (D1 to D4) with varying *φ* (=10, 15, 20, and 25°) and constant *θ*: (**a**) *θ* = 20°, (**b**) *θ* = 0°, and (**c**) *θ* = −45°. The scale bars in the figure and insets in Panel 1 correspond to a 2 µm and 500 nm scale, respectively.

**Figure 3 nanomaterials-12-03464-f003:**
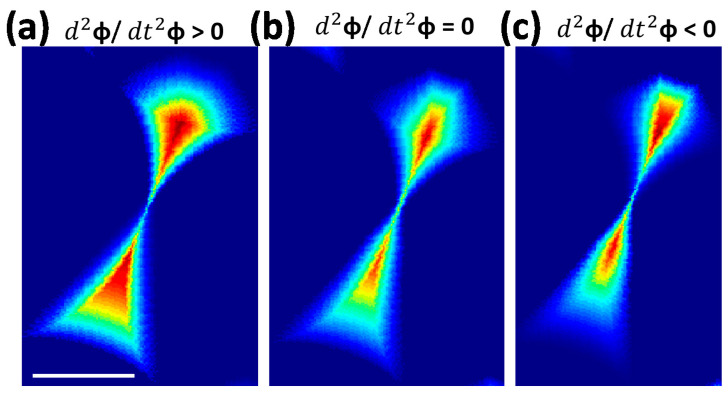
(**a**–**c**) Structures from full rotation of *θ* at different accelerations. Scalebar corresponds to 500 nm.

**Figure 4 nanomaterials-12-03464-f004:**
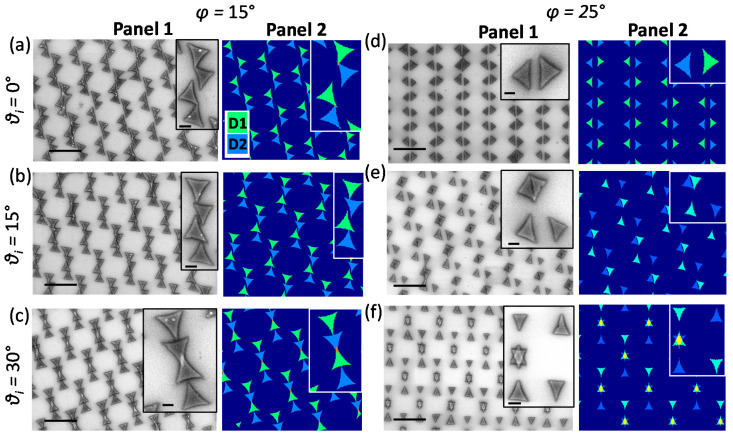
Two-step deposition varying *θ* (*θ_i_* and *θ_i_* + 180°), keeping φ constant: (**a**–**c**) φ = 15° and (**d**–**f**) φ = 25°. The SEM images are shown in Panel 1, whereas the simulation results are shown in Panel 2. D1 and D2 correspond to the two steps of deposition. The scale bars shown in the figure and insets of Panel 1 are of 4 µm and 500 nm, respectively.

**Figure 5 nanomaterials-12-03464-f005:**
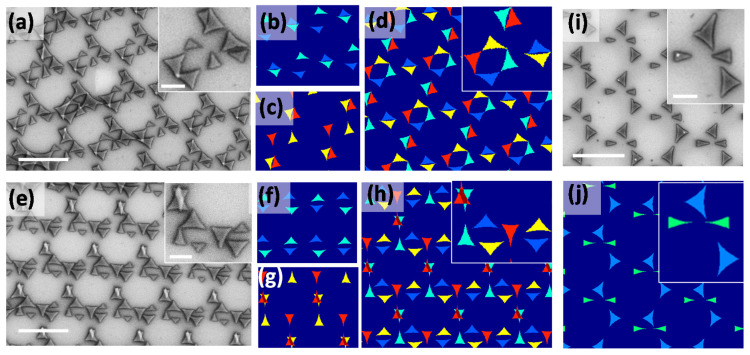
(**a**–**h**) Four-fold θ symmetry structures at constant φ and (**i**,**j**) semi four-fold symmetry. The scale bars in the figure and insets correspond to 2 µm and 500 nm, respectively.

**Figure 6 nanomaterials-12-03464-f006:**
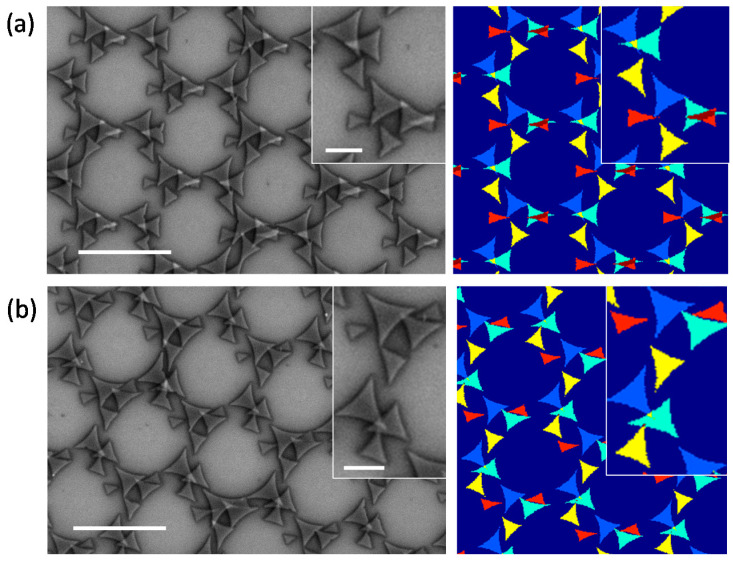
Four-step deposition with simultaneously varying *θ* and *φ*. Insets show the magnified images of unit cells. The scale bars in the figure and insets indicate 2 µm (**a**) and 500 nm (**b**), respectively.

**Figure 7 nanomaterials-12-03464-f007:**
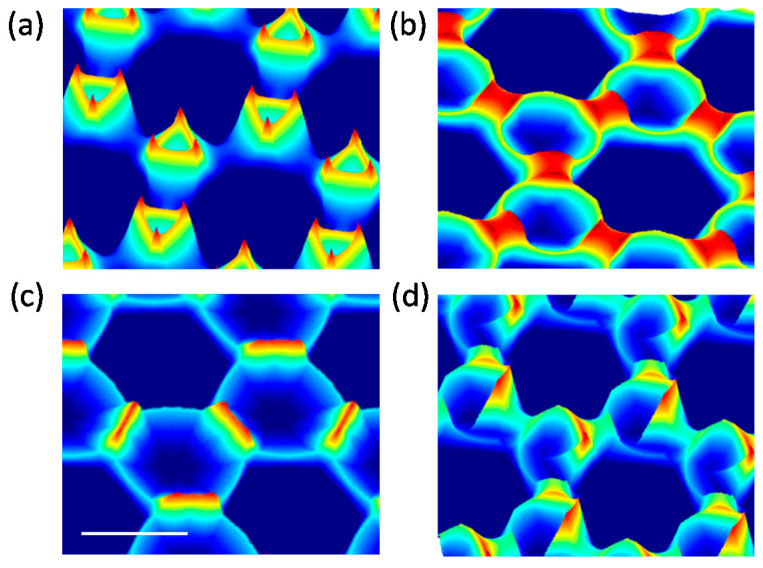
Simulated structures at (**a**–**c**) constant *φ* and (**d**) contentiously varying *φ*, upon full rotation of *θ*. Scalebar correspond to 1 µm.

**Table 1 nanomaterials-12-03464-t001:** Angle configurations of Figure 5.

Figure	Method	*θ*	*φ*
Figure 5a	Experiment	15, 105, 195, 285°	25°
Figure 5b	Simulation	15, 195°	25°
Figure 5c	Simulation	105, 285°	25°
Figure 5d	Simulation	15, 105, 195, 285°	25°
Figure 5e	Experiment	355, 85, 175, 265°	25°
Figure 5f	Simulation	355, 175°	25°
Figure 5g	Simulation	85, 265°	25°
Figure 5h	Simulation	355, 85, 175, 265°	25°
Figure 5i	Experiment	0, 90°	25°
Figure 5j	Simulation	0, 90°	25°

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
