# Peer review of "Combining Azimuthal and Polar Angle Resolved Shadow Mask Deposition and Nanosphere Lithography to Uncover Unique Nano-Crystals"

_nanomaterials, 2022, doi:10.3390/nano12193464_

Round 1

Reviewer 1 Report

In my observation, the concept of this article is well-known. The general fabrication process can be found in the literature. I just give some comments for improving the manuscript before it can be considered for publication in this journal.

-          In the introduction, the potential use of this method for practical application should be reviewed.

-          A scheme to describe the overall experiment and concept should be added.

-          Why does nanosphere lithography in this manuscript only show discrete nanostructures but not continuous structures since space still appears around nanospheres on the mask (Figure 1a.).

-          What is the purpose of this manuscript in fabricating different discrete nanostructures by adjusting the θ and φ. I believe this is the major question that should be addressed. The author should present the property measurements to present the meaning of this study.

-          Shadow deposition should be revised in a better and correct term.

Reviewer 2 Report

Line 88, use Pa instead of Torr

Fig. 3 scalebar missing

Fig. 7. scale bar missing

Round 2

Reviewer 1 Report

The authors have paid much effort to improving the manuscript. As such, I recommend publishing this manuscript in the journal after a minor revision.

- Reference should be corrected in terms of journal name abbreviation and title format.
